# How Select Strength and Power Measures Relate to FCS Football On-Field Performance

**DOI:** 10.3390/jfmk10020193

**Published:** 2025-05-27

**Authors:** Seth Long, Nanette V. Lopez, Jay T. Sutliffe, Dierdra Bycura, Jessica R. Szczepanski, Scott N. Drum

**Affiliations:** Health Sciences Department, Northern Arizona University, Flagstaff, AZ 86011, USA; seth.long@nau.edu (S.L.); nanette.lopez@nau.edu (N.V.L.); jay.sutliffe@nau.edu (J.T.S.); dierdra.bycura@nau.edu (D.B.); jessica.szczepanski@nau.edu (J.R.S.)

**Keywords:** athlete, competition, sport, playmaker, American football, recruit

## Abstract

**Objective:** Understanding unique ways that strength and power contribute to on-field performance in collegiate-based American football might aid coaches in recruiting and determining starters. Using retrospective analysis of existing data, including starting status (STR) and number of defensive tackles or contributing plays (CP), we sought a viable strategy to observe on-field play. Our purpose was to determine what role baseline and in-season strength, and power metrics contributed to on-field football performance (e.g., using STR and CP) from one Football Championship Subdivision (FCS) university. We hypothesized greater pre-season (baseline) and in-season (repeated variables) strength and power outcomes would lead to an increased number of STR among players (*n* = 53) as well as CP among defensive players (*n* = 30). **Method:** Power, determined through countermovement jump (CMJ) was observed weekly using a VALD Performance force deck (i.e., jump height) over a 9-week, in-season period (excluding weeks 6 and 8, a bye week, and erroneous data, respectively). Baseline measures of strength and power were also collected at the beginning of the season for each player using four specific measurements, namely the following: (a) peak vertical jump; (b) pull-ups to failure; (c) a one rep max for bench press; (d) power clean. Pearson’s correlation was used to correlate baseline measures and weekly power, along with baseline measures and the total number of STR and CP each week. Additionally, linear regressions were used to examine the effects of baseline measures (vertical jump, bench press) on STR and CP. **Results:** Moderate correlations (r > 0.5) were observed between baseline variables and weekly CMJ measures. Baseline measures of power clean were correlated with CP only in week 4. All other analyses were not significant. **Conclusions:** Since our on-field performance variables were not significant, future research should focus on more potent variables, as reported in the literature, such as football IQ, initial recruiting status, and psychological resilience, in addition to accounting for strength and power metrics.

## 1. Introduction

More full scholarship opportunities in collegiate football programs [1] vs. other collegiate sports emphasize the importance of the recruiting process. As the college football industry continues to change and grow, coaches seek ways to give their team a winning advantage. This is especially difficult in the new recruiting and retention landscape of yearly transfer portals and name, image, and likeness (NIL) deals. Since the National Collegiate Athletic Association (NCAA) now allows athletes to get paid for their likeness and image, this adds a highly variable and ever-evolving layer to the already difficult decision-making process of athletes, coaches, and universities vying for the best possible outcome and/or quick fix to an existing roster. Many schools utilize their (brand) name, past and current successes, and other opportunities to attract the best high school, junior college, or transfer athletes to their programs [2]. Notably, the most successful programs get the best athletes or recruits because they demonstrably offer the biggest stage for gametime play and personal or monetary exposure [2,3]. In short, an athlete’s loyalty to one team is fast waning. As coaches aim to create successful programs in an unpredictable environment, understanding the impact of more common variables such as strength and power could better inform coaches how to approach these variables when seeking to understand their significances in predicting player performance.

In order to appreciate the way recruiting could evolve, it is important to first understand the recruiting methods presently used. Currently, college football programs utilize athlete-driven video footage of game time performances as well as baseline strength, speed, and power measures to better recognize the potential of an incoming athlete (e.g., recruit) [4]. Various types of testing protocols can be found across different programs and training camps (e.g., summer camps). Performance tests used during training camps usually include the following: (a) 185 lb bench press until failure, (b) vertical jump height, (c) standing broad jump, (d) 40-yard dash, and (e) shuttle run drill [5]. In most circumstances, college coaches recruit athletes who achieve the best outcomes in each of the aforementioned categories along with the athlete’s supposed and evaluated ability to perform on the field [4]. To challenge this narrative, we critically consider select performance-related tests and how they might correlate with greater success on the field as a starter (STR) or select defensive player related to contributing plays (CP, i.e., tackles).

Additionally, when coaches begin the recruiting process, they analyze the athlete’s potential ability to make contributing plays [4]. As alluded to previously, this is usually highlighted in a personalized skill demonstration video that high school and junior college athletes send to prospective coaches. When a football athlete first arrives on a college campus, the goal is to help them improve their individualized athletic ability, including the augmentation of speed, power, strength, and, notably, increase football IQ [2,6,7,8]. According to previous research, strength and power play a dominant role in an athlete’s ability to perform optimally [2,9,10,11,12]. Many collegiate strength coaches require athletes to achieve certain fitness levels because they believe it will optimize performance [13]. For example, in the football program that we evaluated, the head strength coach required incoming athletes to clean 75% of their front squat max, squat 1.6 to 1.8 times their body weight, and achieve 5–15 strict pull-ups based on position.

While there are a variety of factors that contribute to an athlete’s ability to perform on the field, the primary aim of this study was to determine what a “playmaker” might look like based on baseline values of strength and power data, along with examining weekly strength and power data measured in-season. Hence, we focused on different ways that strength and power attributes might contribute to gametime, on-field relevancy. Understanding this could provide college football coaches together with strength and conditioning professionals added insights into how strength and power might predict or facilitate starting status and contributing plays across a season of FCS football.

To better comprehend the impact of these goals on gametime performance, the research team used baseline (pre-season) and in-season, repeated (weekly) strength and power data to test if higher values correlated with starts (STR) and/or contributing plays (CP). We also examined the effects of baseline (i.e., pre-season) strength and power on STR and CP at the end of the season. We hypothesized that athletes with higher baseline strength and power would have a greater number of STR and CP [2,9,10,11,12]. We also hypothesized that baseline strength and power measures would correlate with weekly CP, total CP, and total STR.

## 2. Materials and Methods

### 2.1. The Participants and Measures

“This research was carried out fully in accordance with the ethical standards of the JFMK” [14]. This retrospective study with deidentified data was based on real-world metrics collected by one school’s FCS strength and conditioning staff, per usual pre-season and in-season protocols. According to this specific FCS program, baseline strength and power were assessed at the beginning of the fall football season (i.e., mid-August). The measures helped inform strength coaches about how to progress volume-load recommendations for each athlete over the course of the season. The baseline strength and power measurements included the following: one rep max for (a) power clean and (b) bench press; (c) a vertical jump test; (d) the number of pull-ups an athlete could achieve in one attempt to volitional failure in good form. Collectively, these baseline measures were chosen by the strength and conditioning specialist and were used to indicate both upper body and lower body strength and power. To be clear, the researchers had no control over the standardization of the pre-test measures.

### 2.2. Protocol

Athlete height (cm) and weight (kg) were measured via a stadiometer and digital scale, respectively (Tanita WB-3000 Digital Physicians Scale, Tanita Corp, Tokyo, Japan). After a 10 min full body dynamic warm-up conducted by the strength and conditioning team, and per National Strength and Conditioning Association (NSCA) guidelines [12,15], 1 repetition max (RM) bench press and power clean were conducted to failure. Vertical jump height was assessed using a Vertec™ device (Sports Imports, Worthington, OH, USA) [12]. Next, strict body weight pull-ups to failure were assessed, whereby full arm flexion (“up” phase) with chin over the bar and full arm extension (“down” phase) were observed each rep, per the strength staff’s protocol. Finally, over a 9-week, in-season period, the strength and conditioning team recorded weekly peak power output (PPO, W/kg) for athletes during a countermovement jump (CMJ) using VALD force decks (VALD Performance, Newstead, Queensland, Australia). The force decks are two metal plates that track movement and power through pressure sensors. There were no pre-test conditions or familiarization sessions instituted prior to the first jump. Hence, each jumping session was standardized to one session per-week where all football athletes participated in the jump prior to their morning lifting session. To perform a CMJ on the force decks, an athlete stepped onto each respective plate with their hands on their hips. The athlete then completed a moderately deep squat and jumped as high and fast as possible while maintaining their hands on their hips throughout the movement. This process was repeated three times with a brief (<5 s) pause in between each jump. Five seconds between jumps was a standardized metric utilized by the strength and conditioning staff based on their past experiences and prior use of the VALD system. The force plates measured the PPO (W/kg) of each jump and automatically recorded each attempt. This protocol was standardized for all athletes for every jump and occurred weekly for the entirety of the data collection period. Notably, the highest PPO of the three jumps was used in the correlation analyses.

Game time starts (STR) and contributing plays (CP) were recorded weekly and entered into a Microsoft Excel™ spreadsheet by the team’s statistics manager. These specific variables were examined by the research team because the number of starts could be applied to every player and the number of tackles was a universal performance indicator for a defensive player (note that a similar ‘universal performance indicator’ was not determined for offensive players). In addition, STR is a universal goal for every player on the team, with coaches typically starting their best players to set the team up for success. It was expected that an athlete would start in a game if they had demonstrated preparedness and were likely to make plays on the field. Starting status was recorded as starting that week (1) or not (0), and summed across seven weeks of data collected. This was due to a bye week (week 6) and a data collection error during week 8. CP equates to the amount of tackles a defender makes in a game [16], directly impacting their team’s likelihood of winning [16]. In addition, CP is the only universal goal for all defenders [16]. An offensive variable was not considered because, while they all have a universal goal of helping their teammate score, offensive players’ specific jobs drastically vary. For instance, the job of an offensive lineman is different compared to the job of a quarterback even though their goal of scoring is the same. More specifically, only one offensive player at a time can score (contributing play) while on defense any defender at any given time can make a tackle and/or score (i.e., the focus in the current study was on tackles). The total number of contributing plays were summed across 7 weeks.

### 2.3. Statistical Analysis

Descriptive statistics, including frequencies, percentages, means, and standard deviations were determined. Bivariate correlations between baseline strength variables (power clean, vertical jump, pull-ups, and bench press) and counter movement jump were determined. Multiple linear regressions, conducted separately for total contributing plays and starting status, were used to analyze the data. For both outcomes, baseline variables of vertical jump and bench press were used as predictors. Lastly, we utilized independent sample t-tests to depict differences between baseline power, strength, and the weekly power variable (i.e., CMJ). Using G*Power [17] with *p* < 0.05 for significance and a power of 0.80, we required 31 participants. A *p*-value of 0.05 was considered to be a significant result. All data were analyzed using IBM SPSS Statistics for Windows, Version 27.0 (Chicago, IL, USA).

## 3. Results

The average height, weight, body mass index, and age for starters (*n* = 27) versus non-starters (*n* = 26) are listed in the table below (Table 1). On average, athlete’s strength and power remained stable (see Table 2). A sample size of 30 defensive players was used for evaluating CP and a sample size of 53 offensive and defensive players was used for evaluating STR.

In the following analyses (excluding week 6, bye week, and week 8, with erroneous data), we report player power (as CMJ) on a weekly (i.e., weeks 1–5, 7, 9), in-season basis (see Table 2). Results from independent sample t-tests showed no differences among baseline power, strength, and weekly power variables between starters and non-starters (*p*’s > 0.05). Weekly CMJ measures were moderately correlated (r’s = 0.5 through 0.7) with baseline measures of power (i.e., vertical jump, power clean) and strength (pull-ups to failure, bench press) (see Table 3). Hence, using CMJ (an easy and quick weekly assessment) as a surrogate for the aforementioned variables seemed viable.

The only significant correlation between CP, a defense only, on-field performance indicator (in this study), and baseline measures of power and strength was at week 4. Otherwise, there were no other significant correlations detected (Table 4).

To further explore on-field performance, we summed the starting status (Table 5) for defense players (*n* = 35) only and conducted bivariate correlations with baseline power and strength assessments. No significant (*p* > 0.05) correlations were found. Due to moderate to strong correlations among baseline power and strength variables (r = 0.43 through 0.80), we opted to use vertical jump (power) and bench press (strength) as baseline performance indicators as these two variables had a moderate correlation (r = 0.43). Results from examining linear regression associations among the baseline variables of vertical jump and bench press with total contributing plays (Table 6) and starting status (Table 7) are presented. No statistical significance was observed with these analyses.

## 4. Discussion

With this retrospective analysis of existing data, we evaluated baseline and weekly variables recorded by the head strength and conditioning coach and staff for a Division 1AA football team. The goals were twofold: (1) to examine baseline assessments of power (vertical jump, power clean) and strength (bench press, pull-ups to failure) with weekly assessments of CMJ over a 9-week, in-season period, and (2) to use baseline assessments of power (vertical jump) and strength (bench press) to predict on-field performance via CP and STR. Our overarching aim was to assist coaches in predicting the on-field success of various players. However, only one significant correlation was found between baseline power and strength with CP (week 4) and no significant correlation was found between baseline power and strength with total starts. This was counter to our original hypothesis because we believed greater strength and power at baseline and throughout the season (correlated with weekly measures of CMJ) would lead to significantly greater numbers of STR and CP and therefore provide prediction metrics for coaches. For instance, vertical jump [12,18] and bench press [15] were found to predict on-field play and forecast future elite player status [12]. This was not supported in the current study.

Still, within the literature, numerous research studies [2,9,10,11,12,18,19,20] supported our hypothesis while Kuzmits and Adams [21] and others opposed it. We kept our initial project aims and utilized easily repeatable metrics that were meaningful to coaches (e.g., vertical jump, bench press, CMJ). Concurrently, we recognized that using only strength and power measures at baseline might not lead to better on-field performance. This is because a myriad of factors can contribute to performance beyond our chosen baseline measures. For instance, an evaluation of the processes both NFL and college coaches used to recruit athletes revealed that football IQ [7,22,23], recruiting status [3,19,24,25], and player psychology [25,26,27,28] all contributed to STR and CP. Expectedly, throughout the college season, STR and CP are bound to change on a weekly basis depending on the type of training, readiness to perform, rate of injury, and eligibility status. In this case, we specifically focused on baseline strength and power while also evaluating a weekly analysis of power. We failed to find any significance (except during week 4) when examining the relationship between baseline strength and power measures with contributing plays or total starts, even though these baseline measures were highly correlated with a weekly performance measure (i.e., CMJ). Thus, future studies should examine how assessments of strength and power measured over time using CMJ, a viable proxy for strength and power, may need to integrate additional measures, such as football IQ, fatigue over the course of the season, and strength training plateaus, to more accurately portray how college athletes can be successful at a higher level.

### 4.1. Strength & Power

While baseline values of strength and power did not predict STR and CP in this study, it is still important to note their relationship to performance. Strength and power both served as physical measurements of an athlete’s ability to move a load quickly and efficiently. We hypothesized that a player’s strength and power (at baseline) would directly predict better performance on the field because higher indicators of strength would theoretically enable athletes to overpower their opponent by overcoming a block, making a tackle, breaking a tackle, or throwing a ball farther. Additionally, power would allow an athlete to move quickly through the resistance of another player or allow them to have greater initial speed at the beginning of a play. Other researchers suggested that while many of these things are accurate, general strength and power may not give high-level athletes an advantage [9,11,29]. Hence, to take advantage of strength and power training, coaches must also implement position specific training volumes and movements, and be aware of the time lag between weekly training and on-field gains [4,9,10,12,15,18,19,29]. Additional researchers concurred that measures for athletes who performed better in the forty-yard dash, vertical jump, and shuttle drill were highly positively correlated with overall performance on the field. Additionally, some of this training may include improving overall speed, jump height, and change of direction quickness. Consequently, specificity of training can lead to better performance as it is positively correlated to on-field movements [15]. This suggests that utilizing various types of training, such as multiple sets and different periodized programs [30] to improve results on these tests and the body composition of the athlete can augment team success.

### 4.2. Future Research and Suggested Playmaker Attributes

An additional goal of this research was to explore other contributing factors that play a role in performance with the intent to better appreciate why our results were insignificant. To confirm, our outcomes countered expectations because the hypothesis predicted that players with more strength and power would be more likely to have STR status and a greater number of CP. Understanding this gives researchers the opportunity to re-evaluate the role of baseline strength and power play (when measured continuously) in an athlete’s ability to perform. As alluded previously, a careful evaluation of the literature suggested that while strength and power can contribute to performance indicators such as STR, there are more factors to consider when predicting an athlete’s success [5,13,22]. To offer readers a better understanding of the complex relationship between variables and performance, a more extensive study of the literature suggested that other variables should also be considered to make player performance predictions more reliable. More specifically, these variables included football IQ, recruiting status, prior player statistics, player psychology, and overall understanding of the game [5,13,22]. Specificity of physical variables such as top speed and change of direction were also factors that should/could be considered or developed due to their strong correlation to optimal performance [15]. If the variables from this study were considered along with these additional variables mentioned in the literature, recruiters and coaches might become more informed on the potential of an athlete. It also suggests that current recruiting methods only utilizing strength and power data may be limiting the ability of coaches to predict players’ success on the field.

Ultimately, for coaches to better evaluate their players and understand their potential on the field, they must adopt ways to measure each of these contributing factors, among others. This can particularly give the coaches having no access to higher ranked athletes an advantage by considering each of these various factors during the recruiting process. Notably, results from previous studies suggested that a well-rounded athlete who scores higher in each area may produce better results on the field as opposed to an athlete who dominates in one or two player attributes, such as strength and power.

### 4.3. Limitations

To improve future research regarding strength and power, certain aspects of our study could be further replicated and expanded. Analysis of our data revealed that our repeated variable (i.e., CMJ) may not have been a true reflection of each athletes’ weekly strength and power due to the potential for inconsistent effort as well as technological errors. Data collection also only occurred once a week, leaving little room for confirming results. Testing frequently in a standardized fashion with consistent verbal encouragement by test administrators may provide a more accurate representation of each athlete’s weekly strength and power data, especially when using the CMJ as a proxy for readiness to perform.

Confounding factors are noted with the intention of aiding researchers in future studies. A potential gap in the literature is the comprehensive analysis of time spent watching film and its impact on game-time performance. Lastly, tracking athlete sleep and nutrition habits along with assessing their school schedules might also impact STR and/or CP (i.e., on-field performance or readiness). Future research should employ these types of lifestyle assessments in conjunction with physical or training variables.

## 5. Conclusions

While the goal of this study was to ultimately grasp the relationship between strength and power and success on the field via the number of starts and the volume of contributing plays, an underlying theme began to emerge. The insignificance of strength and power when correlated with STR and CP, found specifically in this study, prompted a more in-depth search into the literature to gain a better understanding of why these variables appeared underwhelming. We found football IQ, recruiting status, player psychology, understanding of the game, top speed, and the ability to change direction as potential singular or collective contributors to on-field performance. Despite our results, strength and power remain noteworthy. For instance, previous studies highlighted the significance of including these two variables as contributors to augmented performance. Appreciating the limitations found in this retrospective study may also suggest the need for the repeated analysis of strength and power to occur in a more controlled environment and systematic manner. This could help aid in the long-term standardization of data tracking and potentially lead to significant outcomes.

In summary, similar studies looking at the relationship between repeated measures of strength and power and performance should be explored to better understand their relationship. It should also be noted that if coaches and recruiters are able to obtain a variety of information from their prospects, they will likely be able to more accurately predict their ability to be successful on the field.

## Figures and Tables

**Table 1 jfmk-10-00193-t001:** Player—starter and non-starter—characteristics.

Status	Number of Players Analyzed per Position	Position	Height, cm (SD)	Weight, kg (SD)	BMI (SD)	Age, yrs (SD)
Starter (*n* = 27)	1	QB	180.3	89.3	27.4	19.0
	1	RB	177.8	111.2	31.9	21
	3	WR	180.7 (2.7)	81.6 (3.7)	25.0 (1.0)	21.3 (0.6)
	1	TE	190.2	113.1	31.3	20.0
	9	OL	192.0 (5.3)	139.4 (11.2)	37.9 (3.5)	21.4 (1.3)
	4	DL	186.8 (3.3)	127.9 (12.8)	36.6 (2.7)	21.3 (1.0)
	3	LB	185.2 (1.6)	108.2 (7.7)	31.5 (2.2)	22.5 (2.1)
	5	DB	182.4 (3.9)	80.1 (24.6)	24.1 (7.4)	21.2 (1.8)
Non-starter (*n* = 26)	2	QB	184.3 (1.6)	90.8 (5.0)	26.8 (1.9)	N/A
	1	RB	182.2	N/A	N/A	N/A
	4	WR	177.7 (5.3)	81.4 (6.7)	25.8 (1.1)	N/A
	4	TE	189.0 (1.2)	110.2 (2.7)	30.8 (0.9)	20.3 (1.7)
	1	OL	193.0	135.1	36.3	21.0
	5	DL	184.7 (0.7)	118.6 (1.5)	34.8 (1.5)	N/A
	5	LB	181.3 (2.0)	97.4 (3.2)	29.7 (1.6)	N/A
	4	DB	179.7 (4.0)	79.7 (5.0)	24.7 (1.1)	N/A

**Table 2 jfmk-10-00193-t002:** Starter and non-starter baseline and weekly assessment outcomes for power and strength.

	Baseline Power		Baseline Strength		Weekly Power: Countermovement Jump (CMJ) (W/Kg)						
	Vertical Jump (m)	1 RM Power Clean (kg)	1 RM Bench Press (kg)	Pull-Ups (# to Failure)	Wk1	Wk2	Wk3	Wk4	Wk5	Wk7	Wk9
Starter	0.7 ± 0.1	128.9 ± 15.0	137.9 ± 20.3	12.0 ± 8.2	55.8 ± 8.3	55.6 ± 8.3	55.4 ± 9.2	53.3 ± 8.8	53.7 ± 9.5	54.0 ± 8.5	52.9 ± 8.1
Non-starter	0.7 ± 0.08	122.4 ± 15.1	134.9 ± 20.7	12.5 ± 6.8	56.8 ± 5.7	55.6 ± 4.9	55.2 ± 5.7	55.4 ± 5.4	56.4 ± 5.2	57.1 ± 5.6	55.7 ± 5.5

Note. During weekly measures, weeks 6 and 8 were excluded due to a bye week and erroneous data, respectively.

**Table 3 jfmk-10-00193-t003:** Bivariate correlations between power (vertical jump, power clean) and strength (bench press, pull-ups to failure) with weekly countermovement jump (CMJ, W/kg).

	Vertical Jump (m)	Power Clean (kg/body wt)	Bench Press (kg/body wt)	Pull-Ups (# to Failure)
CMJ Week 1	0.689 **	0.710 **	0.587 **	0.613 **
CMJ Week 2	0.780 **	0.713 **	0.579 **	0.607 **
CMJ Week 3	0.764 **	0.777 **	0.569 **	0.662 **
CMJ Week 4	0.756 **	0.743 **	0.575 **	0.673 **
CMJ Week 5	0.744 **	0.720 **	0.556 **	0.646 **
CMJ Week 6	0.702 **	0.700 **	0.567 **	0.632 **
CMJ Week 7	0.688 **	0.742 **	0.628 **	0.639 **
CMJ Week 9	0.782 **	0.734 **	0.737 **	0.553 **

Note. Week 6 (a bye week) is included because CMJ was still evaluated., ** *p* < 0.01.

**Table 4 jfmk-10-00193-t004:** Bivariate correlations between baseline power (vertical jump, power clean) and strength (bench press, pull-ups) and contributing plays—defensive players only (*n* = 35), including linebacker (*n* = 8), defensive back (*n* = 17), and defensive lineman (*n* = 10).

	Vertical Jump (m)	Power Clean (kg/body wt)	Bench Press (kg/body wt)	Pull-Ups (# to Failure)
Contributing Plays Week 1	−0.008	0.023	−0.231	−0.182
Contributing Plays Week 2	−0.048	0.352	0.054	0.078
Contributing Plays Week 3	0.268	0.241	−0.193	0.276
Contributing Plays Week 4	0.299	0.610 **	0.183	0.343
Contributing Plays Week 5	−0.089	0.046	0.015	−0.017
Contributing Plays Week 7	0.041	0.184	0.062	−0.116
Contributing Plays Week 9	0.019	0.437	0.078	0.002
Contributing Plays Total	0.093	0.207	0.021	0.028

Note. During weekly measures, weeks 6 and 8 were excluded due to a bye week and erroneous data, respectively. ** *p* < 0.01.

**Table 5 jfmk-10-00193-t005:** Bivariate correlations between baseline power (vertical jump, power clean) and strength (pull-ups, bench press) and sum of starting status—defensive players only (*n* = 35).

	Vertical Jump (m)	Power Clean (kg/body wt)	Bench Press (kg/body wt)	Pull-Ups (# to Failure)
Sum Starting Status	0.086	−0.028	−0.150	−0.035

Note. During weekly measures, weeks 6 and 8 were excluded due to a bye week and erroneous data, respectively.

**Table 6 jfmk-10-00193-t006:** Linear regression associations among baseline variables (vertical jump and bench press) with total contributing plays for defense players only (*n* = 30).

	R^2^	B	Std Error	*p*
Vertical Jump	0.011	0.594	1.137	0.605
Bench Press		0.868	19.582	0.965

**Table 7 jfmk-10-00193-t007:** Linear regression associations among baseline variables (vertical jump and bench press) with starting status with offense and defense players combined (*n* = 53).

	R^2^	B	Std Error	*p*
Vertical Jump	0.019	0.020	0.108	0.856
Bench Press		−2.020	2.132	0.348

## Data Availability

We will provide data upon request.

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
