# Peer review of "How Select Strength and Power Measures Relate to FCS Football On-Field Performance"

_jfmk, 2025, doi:10.3390/jfmk10020193_

Round 1
Reviewer 1 Report
Comments and Suggestions for Authors
The authors are reporting on a retrospective analysis that has sought to better understand the utility and relationship of several off-field training monitoring and adaptation metrics recorded during the pre-season and in-season to broad match performance quality descriptors. In general the writing is suitable although at times it is too conversational which needs improving and the topic will be of interest to readers. The limited use of the in-season repeated measures to only PPO is a significant detriment to the broader application of this manuscript, especially considering that the VALD software can export a significantly greater number of metrics of which a number have been reported to have a strong relationship with on-field performance. Statistically the authors have not made a justification for the approach as the predictor variables do not have a linear ability to continue to increase (improve) so the authors need to justify why the prediction modelling should be linear?
Ln 9 and 14; The use of FB as an abbreviation is unnecessary and does not improve the readability
Ln 12; The authors statement, "... we sought a standardized way to monitor on-field play." is not congruent with the next sentence which is about the relationship of off-field performance monitoring metrics during the pre and in-season to on-field during competition performance. I suggest re-wording or deleting the text.
Ln 14; Change 'predicted' 'to hypothesised
Ln 15; Change 'continuous' to 'repeated' here and throughout the manuscript as the term continuous is applied differently when referring to data.
Ln 60-61 and 138-142; While it can be appreciated that STR covers all players the author decision to not find a way to describe the performance of the offensive positions is a critical limitation to this analysis. This reviewer would have thought that the overall objective of the offensive line was to gain meters, as such the offensive team is all contributing to meters gained per play regardless of position and that the average meters gained per play is an objective metric of their performance.
Ln 85-87; The hypothesis as stated does not cover the entire work described as having a relationship between variables is not the same as predicting an outcome. Please refine.
Ln 98; The authors clearly state that the as the reader may expect in this cohort a measure of lower body strength his performed ie Front squat in this case. However there is no descriptive data, relationship or regression outcomes that report on the use of the front squat
Ln 99; Is the vertical jump tests performed using the same force plate arrangement? If so why is the baseline reported as jump height but the repeated measures in-season reported using peak power output?
Ln 119; This description needs improvement. To be precise the force plates measured the GRF and then the PPO is calculated. However what is then used in the analysis, was then an average of the 3 CMJ PPO's used in the correlation or was it the max PPO from the 3 jumpsused in the correlation analysis?
Ln 123; Change 'continuous' to 'repeated' measures
Ln 150; Please provide a justification for the use of Jump height during baseline CMJ's but PPO during in season repeated measures of CMJ's.
Ln 156; How was data entered to the linear regression, was it step forward or backward approach based on what? How then did the authors consider any time lag effects for possible improvements in repeated measures and when these are actually translated into on-field performance improvements?
Ln 158-159; Glad to see that the authors considered the sample size of the retrospective analysis but on which variable was this sample size calculation performed?
Ln 216-219; Here is an example of writing tone that is too conversational, please edit
Ln 227; Instead of referring to a serial power measure I suggest that the authors are instead referring to repeated off-field performance measures.
Ln 228; Be more precise in the description of the interpretation as the authors limited this relationship to a single derived metric PPO rather than consideration to a range of other metrics that can be used to describe the CMJ performance which may then have given a very different insight to STR status and CP performance.
Ln 248-249; This is clumsily written and with poor grammar.
Ln 249-250; While this reviewer agrees with the assertion it is also why there is the need for a time lag in the analysis for the translation of 'gym' based gains to on-field gains
Ln 286; The authors need to consider other limitations not considered in the model ie sleep duration and quality, other stresses eg exams/asignments, nutrition, etc
Ln 290-292 and 295-298; This reviewer disagrees and believes this is an erroneous assumption by the authors because where then is the 'science' in the sport science being applied to athlete preparation? If an S&C coach is/has been making program decisions based on data collected in this manner then the limitation is that scientific principles are not being followed during data collection, which is a poor reflection of the professional practice being applied.
Ln 294-295; I suggest that the authors reconsider how they are positioning this perceived limitation and instead reflect on the need to seek alternative ways to provide extrinsic feedback as a motivating tool to that athletes would be a better choice of description for the limitation rather than a coach needs to be present.
Ln 299-306; This paragraph has very limited relevance to the data reported and could be much more concisely presented
Ln 354; This Appendix is not referred to in text and is not related to the data reported and as such should be removed
Comments on the Quality of English LanguageSuggestions on the quality of the written English are contained above.
Reviewer 2 Report
Comments and Suggestions for Authors
Dear authors,
I have reviewed your manuscript entitled: "How Select Strength and Power Measures Relate to Division 2 1AA Football On-Field Performance". This study aimed to examine the correlations and predictors between player categories and performance in physical tests throughout the season.
I would like to congratulate you on the research you have carried out. After finalizing the revision of the manuscript, I would like to make a few comments:
Introduction
Must be improved:
The first sentence is a bit vague and could be more direct. Try to be more specific in this chapter.
You should use more specific language and avoid phrases like “unique shapes” — instead, state what you are measuring and why. Improve sentence structure (e.g. break up long or awkward sentences).
Phrases like “from the first author’s personal experience as a D-I football player” reduce scientific rigor. Although interesting, personal experiences are no substitute for empirical data.
Method
The methodology has several flaws:
Phrases like “retrospective analysis of existing data” and “baseline (pre-season)” might be more accurate.
How were height and weight measured? What are the Protocols and Instruments used and respective authors of the Bench Press, Power Clean, CMJ and pull-ups.
It remains to be explained why the height of the jump (in meters and with only one decimal place) is used as an indicator of power. If you used the power variable, the unit of measurement (Watts) and how it was calculated are missing!
Statistics - Small sample size
Although the power analysis justifies this sample size, it is still too small for multiple regression with 4+ predictors. Limited statistical power and increased risk of Type II errors (failure to detect real effects). Furthermore, small samples reduce generalizability.
No multicollinearity check mentioned
“…power clean, vertical jump, push-ups, and bench press were used as predictors.” These variables may be highly correlated (e.g., power clean and vertical jump), leading to multicollinearity. Multicollinearity can distort the true relationship between individual predictors and the outcome.
Results
Be explicit that the correlation was not significant. Avoid passive constructions like “our hypothesis was rejected.”
Discussion
Although the discussion seems honest, adequate and well-intentioned and presents some relevant contributions to the field of sports physical preparation, however, it lacks greater analytical depth and scientific rigor at times. There are good practical reflections, but they could be enriched with more evidence and theoretical structure.
Conclusion
Much of the content of the conclusion repeats aspects included in the discussion. A good conclusion should be more objective and incisive based on the defined objectives and the results achieved.
Round 2
Reviewer 1 Report
Comments and Suggestions for Authors
I thank the authors for their thoughtful responses and engagement in the peer review process and the writing style has been greatly improve.
Specific Comments
Ln 301-302; The paired t-tests are not described in the methods
Author Response
Full Reviewer 1 comment from ROUND 2: I thank the authors for their thoughtful responses and engagement in the peer review process and the writing style has been greatly improve.
Specific Comments
Ln 301-302; The paired t-tests are not described in the methods
Response From Authors: In the most recent manuscript with “track changes” still in place, this is where we find the mention of t-test: Ln 159 “Results from independent samples t-tests showed no differences among baseline power, strength and weekly power variables between starters and non-starters (p’s>.05).”
With the above in mind, Reviewer 1’s comments pertain to Ln 301-302; however, we do not see - in these lines - mention of paired t-tests. Please let us know what we may have missed or misinterpreted about your feedback. Thank you.
Hence, we searched the manuscript and found the only mention of t-tests was when we mentioned the use of what we quoted above about using ‘independent sample t-tests…” in Ln 159.
Finally, we updated the use of “t-tests” in the methods section as the following: “Lastly, we utilized independent sample t-tests to depict differences between baseline power, strength, and the weekly power variable (i.e., CMJ).” This can be found in the latest iteration of this manuscript - attached.